# Abnormal platelet immunophenotypes and percentage of giant platelets in myelodysplastic syndrome: A pilot study

Yi-Feng Wu[1,2,3], Ming-Huei Gu[4], Chao-Zong Liu[5], Wei-Han Huang[1,6], Sung-Chao Chu[1,2], Tso-Fu Wang[1,2]*

**1** Department of Hematology and Oncology, Hualien Tzu Chi Hospital, Buddhist Tzu Chi Medical Foundation, Hualien, Taiwan, ROC, **2** College of Medicine, Tzu-Chi University, Hualien, Taiwan, ROC, **3** Ph.D Program in Pharmacology and Toxicology, School of Medicine, Tzu Chi University, Hualien, Taiwan, ROC, **4** Department of Laboratory Medicine, Hualien Tzu Chi Hospital, Buddhist Tzu Chi Medical Foundation, Hualien, Taiwan, ROC, **5** Department of Pharmacology, School of Medicine, Tzu Chi University, Taiwan, ROC, **6** Department of Clinical Pathology, Hualien Tzu Chi Hospital, Buddhist Tzu Chi Medical Foundation, Taiwan, ROC

* tfwang@tzuchi.com.tw

## Abstract

### Objectives

Myelodysplastic syndrome (MDS) is a heterogeneous hematopoietic stem cell disorder with thrombocytopenia. Flow cytometric immunophenotyping of blood cells has been instrumental in diagnosis as co-criteria, but the data regarding platelets remains lacking. This study aims to determine if there is a difference in surface antigen levels on platelets by comparing surface antigen levels in MDS patients and healthy control subjects. Concurrently, as flow cytometric gating can reveal the diameter of cells, this study will investigate differences in giant platelet percentage by comparing these percentages in high- and low-risk MDS patients.

### Study design

Twenty newly diagnosed MDS patients were enrolled in this study. Platelet surface antigen levels were determined by measuring the binding capacity of antibodies with flow cytometry.

### Results

Platelets of MDS patients were shown to have a lower level of CD61 and higher levels of CD31 and CD36 than healthy controls. Judged by forward scatter (FSC), MDS patients' platelets appeared to be larger than those of healthy control subjects, whereas the MFI adjusted by diameter (MFI/FSC ratio) of CD31, CD41a, CD42a, CD42b and CD61 on platelets were lower in MDS patients than in healthy control subjects. There was a significant quantity of giant platelets found in MDS patients, and the high-risk MDS patients tended to have a higher percentage of giant platelets than low-risk patients. Conclusions: All the results indicate that MDS patients exhibit a lower antigen presentation (MFI) adjusted by diameter on platelets than healthy controls and the giant platelets detected by flow cytometry might correlate with the condition of MDS.

**Data Availability Statement:** The datasets generated during and/or analyzed during the current study are not publicly available due to "Human research participant data" but are available

from all authors on reasonable request. This restriction was from "Institutional Review Board of Tzu-Chi General Hospital". (https://hlm.tzuchi.com.tw/rec/ and https://hlirb.tzuchi-healthcare.org.tw/irb/login) If non-authors need the detail data, please contact with Yi-Feng Wu (wuyifeng43@gmail.com), Ming-Huei Gu (lily0350451@gmail.com) or Tso-Fu Wang (tfwang@tzuchi.com.tw). Because of the asking from the editor, non-author contact information with "Tsung-Chia Yu, Nurse practitioner of department of Hematology and Oncology, Hualien Tzu Chi Hospital, s6011551119@gmail.com" , is added here for a data access committee, ethics committee, or other institutional body to which data requests may be sent.

**Funding:** This study is supported by Hualien Tzu Chi Hospital, Buddhist Tzu Chi Medical Foundation, Taiwan, ROC. The funders had no role in study design, data collection and analysis, decision to publish, or preparation of the manuscript.

**Competing interests:** NO - The authors have declared that no competing interests exist.

## Introduction

Myelodysplastic syndrome (MDS) is a heterogeneous disorder of hematopoietic stem cell characterized by bone marrow failure and an increased risk of acute myeloid leukemia [1]. Patients with MDS usually are elderly and present with anemia, leukopenia, and/or thrombocytopenia [2]. According to 2016 WHO guidelines, the standard diagnosis of MDS remains to be with bone marrow biopsy and aspiration, and the severity of MDS relates to blast percentage [3]. As notable features of MDS include an overlapping with aplastic anemia, paroxysmal nocturnal hemoglobinuria, and other hematologic disorders, diagnosis of MDS remains a challenge; therefore, more refined diagnostic criteria for MDS have recently been proposed [3]. MDS diagnosis is now based on cytopenias, including anemia, neutropenia, and thrombocytopenia; dysplasia by morphologic evidence; and cytogenetic abnormalities [4].

For many years, flow cytometric immunophenotyping has been instrumental in MDS diagnosis as a co-criteria [5] but only immunophenotypic markers on myeloid and monocytes are included. Both the immunophenotypes and DNA ploidy status of megakaryocytic precursors have been investigated in healthy subjects and MDS patients with flow cytometry [6]. However, the analysis needs a large amount of bone marrow (BM) and requires a complicated preparation procedure to prevent aggregation and damage to megakaryocytes [7–10]. As a result, a rapid and feasible immunophenotyping analysis of megakaryocytic cells is necessary for the diagnosis of MDS patients, especially those with refractory thrombocytopenia and multi-lineage dysplasia.

Circulating platelets are offspring of megakaryocytes and inherit a lot of characteristics, including immunophenotypes. Platelets thus appear to be an alternative cellular target for evaluation of megakaryocytic dysplasia. Sandes et al., demonstrated multiple immunophenotypic abnormalities on the platelets of MDS patients, but only half of the patients exhibited altered phenotypes on platelets [11]. It remains unclear whether alterations in the immunophenotypes on platelets are a distinctive characteristic of MDS in comparison to normal individuals. This study aims to determine the levels of antigens on the platelets of MDS patients by flow cytometry and investigate what relevance these levels have MDS.

## Materials and methods

### Participants

Twenty newly diagnosed MDS patients were enrolled in this study. For comparison, twenty healthy adults were enrolled as control subjects. They were recruited at the Buddhist Hualien Tzu-Chi Hospital (Hualien, Taiwan) from January 1st to December 31st, 2012. According to 2007 WHO criteria, diagnoses were made by clinical data, morphological examination from BM smears, and cytogenetic studies [12]. BM smears were subjected to Liu's stain at least 500 nucleated cells were counted in both groups. Erythroid, myeloid, or megakaryocytic dysplasia was defined with more than 10% cells in the BM smear. Patients were categorized into MDS with multi-lineage dysplasia (RCMD, blast less than 5%) (n = 8); MDS with excess of blasts type 1 (EB-1, blasts: 5 to 9%) (n = 3); MDS with excess of blasts type 2 (EB-2, blasts: 10~19%) (n = 1), and MDS-t (MDS with acute leukemic transformation or RAEB-t, blasts: 20~30%) (n = 8). None of the 40 participants took any medicine that wound affect platelet function or production during the two weeks preceding the start of the study.

This study was approved by the Institutional Review Board of Tzu-Chi General Hospital (IRB101-119). Informed consent was obtained after the explanation of the study protocol.

## Blood sample preparation

After enrollment, peripheral blood was taken from MDS patients before receiving treatment. None of the patients received platelets transfusion or target therapy, included Azacitidine or Decitabine, before drawing of blood samples for flow cytometry. Samples were collected in EDTA-containing tubes by venipuncture and kept at room temperature before analysis. EDTA was employed as the anticoagulant because it is a strong divalent chelator capable of inhibiting platelet activation and aggregation, as well as platelet-leukocyte aggregation. Around 5 mL of whole blood was collected for complete blood count (CBC) and flow cytometric analysis.

## Complete blood count test

The complete blood count (CBC) test was carried out with a Sysmex XE-5000 hematology analyzer (Sysmex, Kobe, Japan). Data regarding white blood cells (WBC), hemoglobin, hematocrit, and platelets were collected. The PLT-O (fluorescence) channel of Sysmex instrument was used for this study. Residual whole blood was diluted with saline (0.9% NaCl) to desired platelet count (10,000/μL) by the data of CBC before flow cytometric analysis. As the platelet count of MDS patients varies among individuals and appears to be lower than that of healthy controls, flow cytometric analysis was performed with diluted whole blood containing the same platelet count.

## Flow cytometry

Flow cytometry was performed within thirty minutes after drawing blood samples, and the platelets were not stored. The level of surface antigens on platelets was determined by the binding capacity of antibodies detected by flow cytometry. An aliquot (45 μL) of diluted whole blood was incubated with a fixed volume (10 μL) of fluorochrome-conjugated antibodies at room temperature for 20 min. The mixture was then diluted with 1 mL of phosphate-buffered saline (PBS) and immediately subjected to flow cytometric analysis by FACSCalibur (Becton-Dickinson Immunocytometry Systems, San Jose, CA, USA). Based on single-cell imaging, flow cytometry allows us to identify specific blood cell types through an appropriate gating on the light scattering profile (FSC vs. SSC) (S1A Fig), and reveal platelets by positivity of the specific markers (S1B Fig). Data were acquired and analyzed by WinList (Verity Software, Topsham, ME, USA). The fluorochrome (FITC or PE)-conjugated antibodies, including TLD-3A12 (CD31-PE) for PECAM-1, CB38 (CD36-FITC) for glycoprotein IV, HIP8 (CD41a-FITC) for glycoprotein IIb, ALMA.16 (CD42a-FTIC) for glycoprotein IX, HIP1 (CD42b-PE) for glycoprotein Ib, and VI-PL2 (CD61-FITC) for glycoprotein IIIa, were purchased from BD PharmingenTM (BD Biosciences, Franklin Lakes, NJ, USA).

In flow cytometric analysis, cell surface antigens can be easily identified with the binding of specific antibodies. As to the level of surface antigens, the data regarding the binding capacity of antibodies is indispensable. In this study, the binding of anti-bodies to platelets was determined by flow cytometry using a direct-staining protocol as previously described [13, 14]. Sample dilution was employed to reduce the back-ground fluorescence intensity. Fluorochrome (FITC or PE)-conjugated antibodies, including CD31-PE, CD36-FITC, CD41a-FITC, CD42a-FTIC, CD42b-PE, and CD61-FITC, were bound to platelets of healthy controls in a dose-dependent manner and reached a plateau at the dosage of 10 μL, as indicated by the mean fluorescence intensity (MFI). S2 Fig shows the binding curve of CD61-FITC as an example. We thus used this dosage of antibodies to determine the level of platelet surface antigens.

## Statistical analysis

GraphPad Prism (Version 7) was used for the analysis of enrolled data. Participants' characteristics, CBC, and platelets' surface antigens were compared between the two groups by T-Test. All statistical data were expressed as the means and 95% confidence interval (95% CI). A P-value <0.05 was considered statistically significant.

## Results

### Demographics and CBC data

The demographics and lab data for patients with MDS and healthy control subjects were shown in Table 1. The distribution of sex between MDS patients and healthy controls was different. In MDS patients, the male: female was 15:5, and the healthy control was 8:12. Age distribution was also different. MDS patients were significantly older (69.5 ± 16.2 years) than healthy controls (35.4 ± 12.2 years) and exhibited a lower hemoglobin level (8.36 ± 1.65 vs. 13.6 ± 1.19 g/dL, p<0.01) and lower platelet count (58.5 ± 80.4 $*10^9$/L and 248 ± 54.9 $*10^9$/L, p<0.01). There were no significant differences in white blood cell (WBC) count and mean platelet volume (MPV) between MDS patients and healthy controls. However, there were 13 patients without MPV values. Accurate measurements could not be obtained due to the presence of red blood cell fragments and giant platelets present in those patients' blood samples. Most MDS patients' peripheral blood contained blast cells with a mean percentage of 5.4 ± 8.2%. By contrast, no blast cells were identified in healthy controls. From the patients' medical records of bone marrow smears, all patients had dysplastic changes over megakaryocytes, including micro-megakaryocytes and separate nuclei. Additional bone marrow smear analysis revealed the numbers of megakaryocytes decreasing in 10 patients, and increasing in 6 patients, while 4 patients maintained adequate megakaryocyte numbers.

From the medical records, symptoms of bleeding (including petechiae, ecchymosis, nasal bleeding, and gum bleeding) were found in 5 MDS patients, each of whom had platelet counts less than 10,000/μL while they were enrolled in this study. No thrombosis events were noted in any of the patients.

### The levels of CD31, CD36, CD41a, CD42a, CD42b, and CD61 on the platelets of MDS patients and healthy controls

The platelets of MDS patients expressed lower level of CD61 (p<0.01) and higher levels of CD31 (p<0.05) and CD36 (p<0.01) than that of healthy controls. There was no significant

**Table 1. Lab data for MDS patients and healthy controls.**

|  | MDS patients (N = 20) | Healthy Controls (N = 20) | P-value |
|---|---|---|---|
| **Sex (Male: Female)** | **15: 5** | **8: 12** | **0.025*** |
| White blood count ($*10^9$/L) | 5.38±5.42 (3.01 ~ 7.76) | 7.00±1.59 (6.30 ~ 7.70) | 0.208 |
| Hemoglobin (g/dL) | 8.36±1.65 (7.63 ~ 9.08) | 13.6±1.19 (13.1 ~ 14.1) | <0.01** |
| Platelet count ($*10^9$/L) | 58.5±80.4 (23.2 ~ 93.7) | 248±54.9 (224 ~ 272) | <0.01** |
| MPV (fl)# | 9.83±1.05 (9.37 ~ 10.3) | 10.1±0.86 (9.74 ~ 10.5) | 0.473 |
| Blast (%) in peripheral blood | 5.4±8.2 (1.47 ~ 9.33) | 0 | - |

Data were shown in mean±SD (95% CI).

The difference between MDS patients and healthy controls was examined by unpaired t-test.

* P-value < 0.05.

** P-value < 0.01.

# There were 13 patients without MPV data.

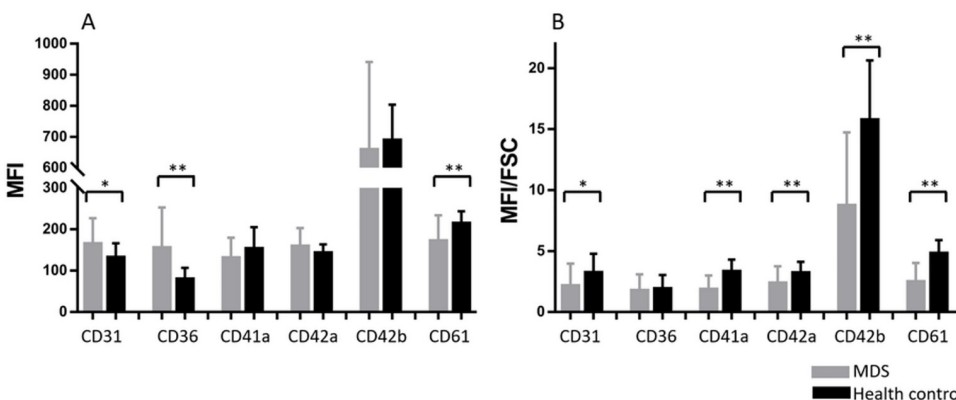

**Fig 1.** (A) The level (MFI) of CD31, CD36, CD41a, CD42a, CD42b and CD61 on the platelets of MDS patients (MDS, n = 20) and healthy controls (HC, n = 20). (B) The MFI adjusted by diameter (MFI/FSC) of CD31, CD36, CD41a, CD42a, CD42b and CD61 on the platelets of MDS patients (MDS, n = 20) and healthy controls (HC, n = 20). Data were shown in mean±SD. An unpaired t-test was employed to examine the difference between MDS patients and healthy controls. * P-value <0.05; ** P-value <0.001.

difference in the levels of CD41a, CD42a, and CD42b between MDS patients and healthy control subjects (Fig 1A and Table 2). From these inconsistent results, each antigen on platelets seems unlikely to relate the same pattern during evolving dysplastic changes.

**Table 2. Immunophenotypic characteristics of platelets in patients with myelodysplastic syndrome and healthy controls.**

|  | MDS patients | Healthy Controls | P-value$ |
|---|---|---|---|
| Antigen level (MFI) |  |  |  |
| CD31 | 164.67±62.33 (137.35 ~ 191.99) | 131.40±34.76 (116.16 ~ 146.63) | 0.044* |
| CD36 | 154.60±98.26 (111.54 ~ 197.67) | 79.06±27.69 (66.92 ~ 91.20) | <0.01** |
| CD41a | 130.50±49.47 (108.82 ~ 152.18) | 152.48±52.82 (129.33 ~ 175.62) | 0.188 |
| CD42a | 158.40±44.45 (138.91 ~ 177.88) | 142.5±21.20 (133.21 ~ 151.79) | 0.157 |
| CD42b | 658.19±283.42 (533.97 ~ 782.40) | 688.58±115.11 (638.13 ~ 739.02) | 0.659 |
| CD61 | 171.45±62.32 (144.14 ~ 198.76) | 213.71±29.89 (200.62 ~ 226.81) | <0.01** |
| Size of platelets |  |  |  |
| FSC | 85.72±40.87 (67.81 ~ 103.63) | 46.71±11.53 (41.66 ~ 51.76) | <0.01** |
| MFI adjusted by diameter (MFI/FSC) |  |  |  |
| CD31 | 2.14±1.82 (1.34 ~ 2.94) | 3.23±1.56 (2.54 ~ 3.91) | 0.05* |
| CD36 | 1.75±1.34 (1.16 ~ 2.33) | 1.90±1.14 (1.40 ~ 2.40) | 0.705 |
| CD41a | 1.85±1.16 (1.34 ~ 2.35) | 3.30±1.00 (2.87 ~ 3.74) | <0.01** |
| CD42a | 2.36±1.39 (1.75 ~ 2.97) | 3.20±0.92 (2.80 ~ 3.61) | 0.03*a5 |
| CD42b | 8.72±6.00 (6.09 ~ 11.35) | 15.74±4.88 (13.61 ~ 17.88) | <0.01** |
| CD61 | 2.47±1.57 (1.78 ~ 3.15) | 4.79±1.10 (4.31 ~ 5.27) | <0.01** |

Mean±SD (95% CI); MFI, Median Fluorescent Intensity; FSC, Forward scattering

$ P-value by T-test

* P-value < 0.05

** P-value <0.01

### The surface antigen MFI adjusted by the diameter of CD31, CD36, CD41a, CD42a, CD42b, and CD61 on the platelets of MDS patients and healthy controls

In addition to the changes of platelet surface antigen levels in MDS, we next sought to examine if there was an alteration in surface antigen MFI adjusted by the diameter of platelets. In flow cytometry, forward scatter (FSC) represents the size of cells. We thus estimated the MFI adjusted by the diameter of surface platelet antigen by dividing antigen level (MFI) by the size of platelets (FSC). MDS patients' platelets appeared to be larger than healthy controls (85.72 ±40.87 vs. 46.71±11.53, p<0.01), whereas the MFI adjusted by diameter (MFI/FSC ratio) of CD31, CD41a, CD42a, CD42b, and CD61 on platelets were shown to be lower in MDS patients (Fig 1B).

### Giant platelets in the peripheral blood of MDS patients and healthy controls

Common platelet size was 3 to 4 μm in diameter. Giant platelets, defined by a diameter of more than 7 μm, has been demonstrated in MDS patients' peripheral blood and BM [15]. They are usually larger than red blood cells. On gating platelets by the surface marker of CD61, a subgroup of platelets bearing exceptionally large size (FSC>1000) similar to WBC and RBC was noticed in MDS patients (Fig 2A), suggesting that these platelets with exceptionally high FSC were giant platelets. We gated all platelets and the platelets with FSC>1000. The percentage of giant platelets was calculated by FSC>1000 platelets counts / total platelets count (gated by flow cytometry). Healthy controls exhibited only trace amounts of giant platelets (Fig 2B). The percentage of giant platelets for MDS patients and healthy controls were calculated to be 7.48 ± 5.35 and 0.74 ± 0.30, respectively.

The MDS patients in this study comprised four of those seven subtypes based on the WHO system, including RCMD, EB-1, EB-2, and MDS-t. RCMD and EB-1 were considered low-risk MDS, while EB-2 and MDS-t belonged to high-risk MDS. There were 9 patients with high risk and 11 with low risk. The high-risk MDS patients tend to have a higher percentage of giant

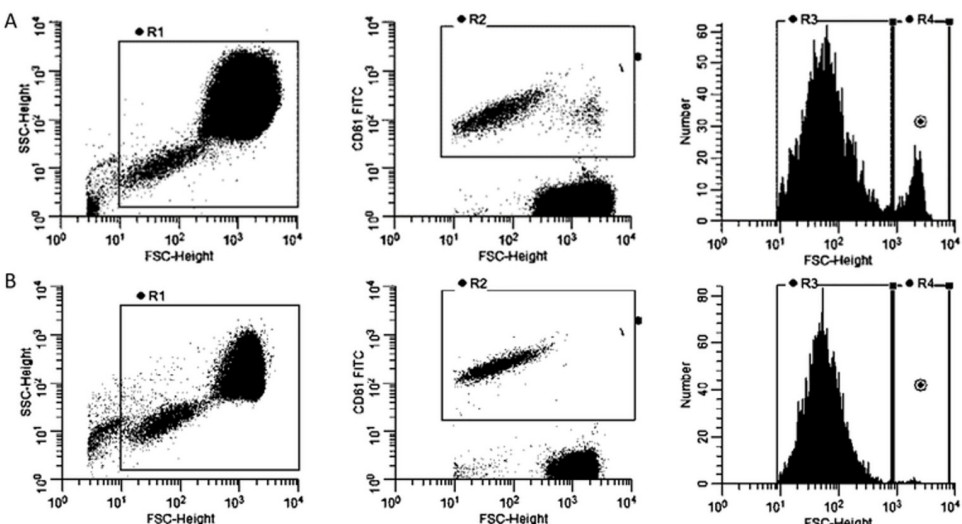

**Fig 2.** (A) The gating of flow cytometry with platelets from an MDS patient. (B) The gating of flow cytometry with platelets from a healthy control.

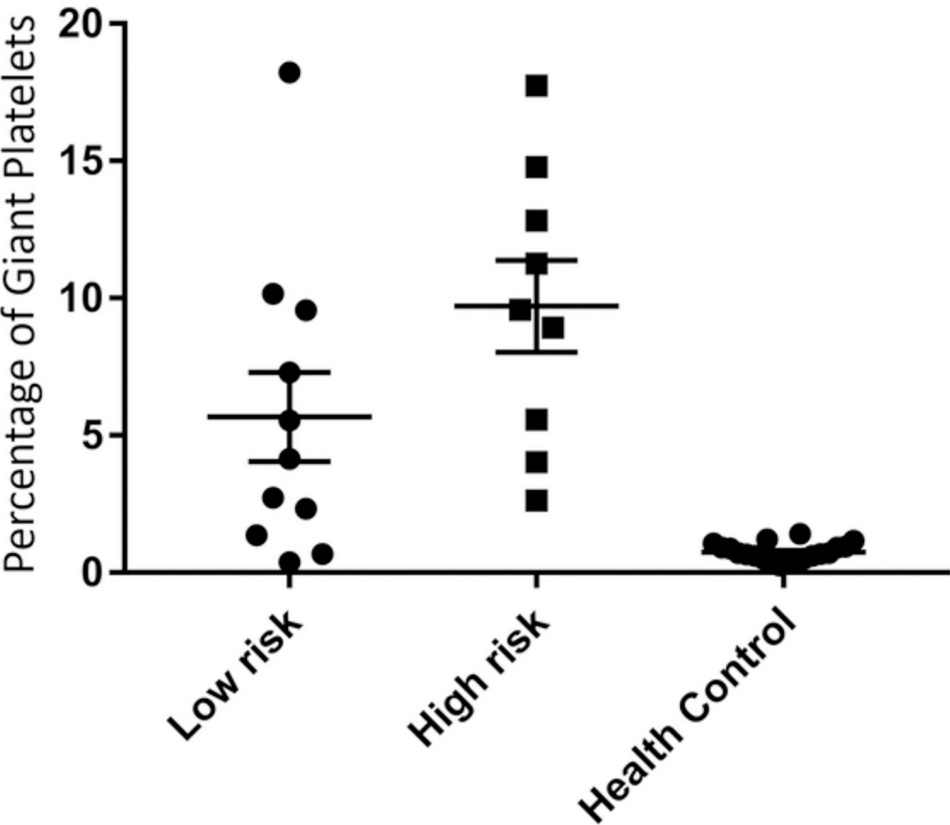

**Fig 3. The percentage of giant platelets in healthy controls and patients with low-risk and high-risk MDS.**

platelets than low-risk MDS patients did (9.70 ± 4.75 vs. 5.67 ± 5.13) (Fig 3). How-ever, the difference failed to reach any statistical significance (p = 0.08). We also calculated the IPSS-R score for our patients. The IPSS-R score was correlated with low- and high-risk, as in our previous subgroups. Nine patients with high risk had IPSS-R score of 5.5–9, and 11 patients with low risk had IPSS-R scores of 1–4.5. High IPSS-R score patients also have higher centage of giant platelet but not significantly different.

## Discussion

Myelodysplastic syndromes are characterized by hemopoietic insufficiency with peripheral cytopenia and an increased risk of leukemic transformation. Conventional diagnostic criteria for MDS are based on morphology and cytogenetics with the exclusion of other disorders which can also cause cytopenia or dysplasia [2]. Multiparameter flow cytometry immunophe-notyping was recently reported to be a useful diagnostic tool for MDS. Data regarding the myeloid, monocyte [16], and erythroid cells [17] had been used for diagnosis and prognostic assessment of MDS, but only a few studies on the megakaryocytic lineage cells have been reported [18]. Since the abnormality of platelets, such as thrombocytopenia, is one of the MDS risk factors associated with worse out-comes [19, 20], it was anticipated that the immunophe-notypic characteristics of platelets would be helpful for the diagnosis and prognosis of MDS.

In this study, flow cytometry was employed to investigate the immunophenotypes of plate-lets for patients with MDS. Whole blood flow cytometry uses only minuscule volumes and applies even to samples with low platelet counts [21, 22].

Megakaryocytopoiesis involves the hematopoietic stem cells to megakaryocyte progenitors, the maturation of megakaryocytes, and the production of platelets [23]. The limitation of bone marrow aspiration is due to the bone marrow sample unavailability from the bilateral pelvic bone. If the abnormal megakaryocytes are focal aggregation, the immunophenotypes of megakaryocytes will show typical results [24]. The flow cytometry of platelets from peripheral blood will overcome this limitation.

Because MDS is a heterogeneous disease and the stage of dysplastic changing of megakaryocytes is not consistent, the changing of platelets' immunophenotypes will not be constant. In our study, we only found a high-burden expression of antigens, including CD41, CD42a, CD42b, and CD61, had significantly decreased levels in MDS patients compared to healthy donors.

Tracing preview papers, the most extensive series for platelet marker of MDS patients was reported by Sandes AF et al. In that study, blood samples from 44 MDS patients versus samples from healthy patients with normal peripheral platelets were compared to identify alterations that could contribute to the diagnosis of megakaryocytic dysplasia in MDS. But the results were not synchronous [11]. Therefore, our study investigated the immunophenotypic profiles of peripheral blood platelets from MDS patients compared with healthy subjects, and we only chose platelet-specific antigens for this study. CD41a and CD61 are the main functions of platelet aggregation. CD42a and CD42b are the second most common platelet receptors while the GPIb-IX-V complex is vital for platelet adhesion. According to previous papers, CD41/ CD61 expressions are induced at the early stage. CD42a/CD42b are later markers for differentiation, preceded by CD61 [25]. Dhanjal et al. presented a key role for CD31 (also named PECAM-1) in regulating the rate and direction of migration of megakaryocytes. CD31 has been shown to regulate megakaryocytopoiesis but is ex-pressed at low levels on platelets. Low levels may cause a non-significant difference in our study [26, 27]. CD36 is expressed following platelet stimulation and plays a role in mediating platelet-platelet and platelet-monocyte cell adhesion as well as cell attachment to the matrix. Previous reports also describe the expression of CD36 in megakaryocyte lineage leukemias, but the stage of megakaryocytes' maturation is still unclear [28]. Increasing CD36 may be caused by abnormal megakaryocyte maturation. As shown in a previously published study by Alex F. Sandes et al, the results showed that 27% of MDS patients had an increased expression of CD36 [11]. Because antigen saturation is vital for evaluating mean fluorescent intensity, and there were no detailed descriptions of the use of binding curves in the previous study, we, therefore, arranged prior research using binding curves for antigen saturation of all platelet surface markers. As noted, our results showed significantly decreasing immunophenotypic characteristics of platelets in MDS patients after adjusting FSC.

Based on the overall immunophenotypic profile of peripheral blood platelets, correlations between the severity of diseases and immunophenotypic characteristics were also investigated. The previous data from Sandes AF et al. using immunophenotypic scores, showed the patients with unilineage dysplasia (RA and RARS) compared with multilineage involvement already among these subgroups of MDS patients [11]. Immunophenotypic platelet score showed a crucial prognostic impact on the outcome of MDS patients in the previous study. In a study by of Jiaxi Liu et al., the data showed FSC, SSC, and MFI of CD41a did not reveal a significant difference between MDS patients and healthy subjects or among different subgroups of MDS [29]. Differences in methods noted in previous studies may have caused different results in our study. First, we used EDTA as an anticoagulant for the prevention of platelet activation and aggregation, as well as platelet-leukocyte aggregation. Although citrate was suggested as an anti-coagulants for platelet study, we used EDTA as a stronger inhibitor for the prevention of platelet activation prior to analysis. Second, we used only one fluoresce channel in each tube to

prevent multi-channels of fluorescence from interfering with each, thus eliminating the need for an adjusting compensation. Third, antigen saturation is very important for MFI. Therefore, before our study, we re-worked the saturation curve for all antibodies. As an additional method, an adequate platelet count was needed to dilute to $10 * 10^9$/L, and 45 μL diluted whole blood was added with 10 μL of fluorochrome-conjugated antibodies. These were found to be sufficient.

Except above, no other studies evaluated the correlation between antigen presentations and the severity of MDS. Our study tried to separate our MDS patients into two groups, high and low risk, according to 2007 WHO classification. We used the FSC to adjust the MFI data of subgroup MDS patients. We found only the percentage of large-size platelets correlated with the severity of MDS, not the MFI adjusted by diameter of platelet surface markers. Despite that high-risk MDS patients are inclined to have a higher percentage of giant platelets than the patients with low risk (Fig 3), statistical analysis was unable to demonstrate a significant difference (p = 0.08). This result might be due to the small sample size of MDS patients and healthy subjects enrolled in this study. A small sample size may result in an insufficient power to detect the difference between groups, even if differences exist in reality. According to previous reports of blood smears from MDS patients, giant platelets are noted with typical morphological abnormalities in MDS cases [30], but when using conventional laboratory routines platelet counts are checked by automated impedance, while typically ignoring large-size platelets. In this study, we could find the large-size group by flow cytometry, even in blood samples with low platelet count. Large-size platelets may play an important role in gauging the severity of MDS in patients and in formulating prognoses.

According to previous publications, MDS subtypes classified by blast percentages had reported a statistically significant difference in overall survival and prognosis [31–33]. In our study, because the percentage of large platelets related to the severity of MDS, it may correlate to the prognosis of MDS patients.

In our study, although the sex ratios between groups were significantly different, no significant differences were noted between males and females in the percentage of giant platelet, FSC, and antigen presentations. In comparing male and female subjects, the P-value was 0.21 in the percentage of giant platelets, 0.28 in FSC, 0.11~0.94 in antigen presentations with MFI, and 0.31~0.85 in MFI adjusted by diameter. Under the published data by Oliver Heidmann Pedersen et al., no difference in platelet aggregation was found between sex and age groups in a cohort of 171 Healthy Individuals with 87 men and 84 women with ages ranging from 21–65years [34, 35]. Whether there are purely age-related changes in platelet surface markers is unclear. We also analyzed the correlation between clinical data and the results of platelet flow. The megakaryocytes' numbers by bone marrow smear showed no significant differences in the percentage of giant platelets, FSC, and antigen presentations. We analyzed the results between 5 patients with bleeding tendencies and other 15 patients. No significant differences were noted between the two groups in the percentage of giant platelet, FSC and antigen presentations. We also analyzed the correlation between platelet counts and the percentage of giant platelets in MDS patients. The R2 = 0.2328. In our data, no correlation was found between platelet counts and the percentage of giant platelets because of the small number of patients.

Turnover of platelets in MDS patients became more important, but almost correlating data was found in chronic myeloproliferative neoplasm with essential thrombocytosis and immune thrombocytopenia. There were few results in myelodysplastic syndrome. Platelet turnover was assessed by immature platelet count, immature platelet fraction and MPV in essential thrombocythaemia. The patients had an accelerated platelet turnover leading to a higher proportion of newly produced immature platelets, and they also had significantly higher expression of platelet granule makers, P-selectin and CD63 [36–38]. From one review article, large platelets

contain higher absolute amounts of proteins, including glycoprotein Ia/IIa and IIb/IIIa complexes, and glycoprotein VI. Although large platelets express 30 to 50% more glycoprotein (GP) Ia, GPIb, and GPIIIa on their membranes compared with small platelets, functional differences between large and small platelets remain poorly understood because of a lack of standardized protocols separating platelets of different size [39, 40]. There was only one study that used autologous 111In-labeled platelets for platelet kinetics and the sites of platelet destruction in five thrombocytopenic patients with myelodysplastic syndromes. The patients with myelodysplastic syndromes had a normal pattern of platelet destruction, and recovery and survival were similar between normal subjects and myelodysplastic patients [41]. By the above study, where there is a higher turnover rate of platelets there may be larger platelets and more glycoprotein receptors or antigen presentations. In our study, for the platelets of myelodysplastic syndrome, giant platelets and lower antigen presentations were found.

The main limitation of our study was the comparatively small groups of patients. Nothing this limitation in our pilot study, we will enroll more patients in further studies to confirm our results. Secondly, the age range in healthy control subjects was younger than the age range in MDS patients. Only one study compared the markers of platelet microparticles by flow cytometry in younger and older patients. The results showed no significant difference in CD41 platelet microparticles between young and elderly patients [42], but no data showed the difference in platelet surface markers by age before. Therefore, the recruitment of healthy control subjects whose age range more closely approaches that of the MDS patients' group will continue to be a strongly desired goal.

## Conclusions

The present study shows the presence of lower platelet immunophenotypes by flow cytometry in MDS patients. From a clinical point of view, assessment of aberrant antigen expression profiles may help in the diagnosis of MDS. And, the FSC was higher in MDS patients with a correlation to the severity of MDS condition. Further study with a large sample size is needed to confirm our results.

## Supporting information

**S1 Fig.** (A) A light scatter profile (FSC vs SSC) of diluted whole blood, in which blood cells are indicated in the rectangular region R1 (B) A histogram showing the blood cells with CD61 positivity.
(TIF)

**S2 Fig. The mean fluorescence intensity of platelets after incubation of diluted whole blood with various doses of FITC-conjugated anti-CD61 antibody.** Data were shown in mean±SD (n = 10).
(TIF)

## Author Contributions

**Conceptualization:** Yi-Feng Wu.

**Data curation:** Wei-Han Huang, Sung-Chao Chu.

**Formal analysis:** Yi-Feng Wu, Chao-Zong Liu.

**Methodology:** Ming-Huei Gu, Chao-Zong Liu.

**Supervision:** Tso-Fu Wang.

**Writing – original draft:** Yi-Feng Wu.

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
