## [Decision Letter · Decision Letter 0]

3 Jun 2022

PONE-D-22-10816Abnormal platelet immunophenotypes and percentage of giant platelets in myelodysplastic syndrome: a pilot studyPLOS ONE

Dear Dr. Wang,

Thank you for submitting your manuscript to PLOS ONE. After careful consideration, we feel that it has merit but does not fully meet PLOS ONE’s publication criteria as it currently stands. We apologise for slight delay on decision . Overall the reviewers felt the report had value. However, a number of important issues were raised that would preclude publication in its current form. All of the reviewers' comments must be addressed in detail for further consideration.Therefore, we invite you to submit a revised version of the manuscript that addresses all points raised during the review process.

We look forward to receiving your revised manuscript.

Kind regards,

Daniel Thomas, MD

Academic Editor

PLOS ONE

Journal Requirements:

“NO - The funders had no role in study design, data collection and analysis, decision to publish, or preparation of the manuscript.”

Reviewers' comments:

Reviewer's Responses to Questions

**Comments to the Author**

1. Is the manuscript technically sound, and do the data support the conclusions?

Reviewer #1: Yes

Reviewer #2: No

Reviewer #3: Partly

2. Has the statistical analysis been performed appropriately and rigorously? 

Reviewer #1: Yes

Reviewer #2: I Don't Know

Reviewer #3: Yes

3. Have the authors made all data underlying the findings in their manuscript fully available?

Reviewer #1: Yes

Reviewer #2: Yes

Reviewer #3: Yes

4. Is the manuscript presented in an intelligible fashion and written in standard English?

Reviewer #1: No

Reviewer #2: Yes

Reviewer #3: Yes

5. Review Comments to the Author

Reviewer #1: Abnormal platelet immunophenotypes and percentage of giant platelets in myelodysplastic syndrome: a pilot study is a well written article. Clinical significance of decreased platelet antigen density in MDS patients might provide more valuable information to readers. There are some comments.

1.The difference of size and platelet antigen expression by sex should be noted among MDS and healthy control group.

2. Density is calculated by mass / volume. The “density” in this article was defined as MFI / FSC ratio which is actually signal value / diameter. The term used in this article “density” might be changed to other terminology, such as signal to diameter ratio or MFI adjusted by diameter, etc.

3. What was the relationship between dysmorphism of megakaryocyte in bone marrow aspiration or biopsy and platelet size or antigen density at the time of blood collection or flow cytometric analysis?

4. Was platelet size or antigen density related with bleeding tendency or thrombus generation?

5. Was platelet size or antigen density related with survival of patients?

6. Was patients transfused with platelet product before blood drawn?

7. Was platelet size or antigen density affected transfusion frequency?

8. Is additional platelet related hematologic parameters available such as PDW, P-LCR, PCT that is provided by XE-5000?

9. Giant platelets in peripheral blood is known to be present in high turnover state (ITP, MPN, MDS….) and falsely decrease platelet count. How was platelet count and ratio of giant platelet associated?

10. What is the clinical significance of measuring giant platelet, which is already known to be associated with high turn over state and of measuring antigen density?

Reviewer #2: Manuscript Number: PONE-D-22-10816

Manuscript Title: Abnormal platelet immunophenotypes and percentage of giant platelets in myelodysplastic syndrome: a pilot study

The authors have studied platelets in blood samples from 20 patients with myelodysplastic syndrome and 20 healthy controls collected in 2012. They aimed to compare the density of antigen expression of platelet-associated CD31, CD36, CD41a, CD42a, CD42b and CD61 between these two groups using standard flow cytometry. Platelets were identified by light scatter (forward; side) and then assessed for expression of the specific molecule. There are a number of issues to be addressed to validate the findings and before the authors claim that this work “may help the diagnose MDS” [sic] can be sustained. These include technical and methodological.

There are discrepancies between the conclusions drawn in different parts of the manuscript. In the Abstract they conclude that in myelodysplasia there is lower antigen density of CD31, CD41a, CD42a, CD42b and CD61 on platelets. In the Results they state there is higher CD31 and CD36 antigen density, lower CD61 and no difference between MDS and normals for CD41a, CD42a and CD42b. This needs explaining.

Patients: Samples were from 20 patients with MDS and the WHO 2006 classification was used. The authors should update these to WHO 2017. Why were indolent types of MDS not included (e.g. single lineage dysplasia). The blood count units should be in international units.

The controls were not age-matched. What changes occur in the density of the antigens studied with normal healthy ageing?

MDS patients had blast cells in their bone marrow (5.4 +/- 8.2%) and healthy controls 0%. Did controls have bone marrow examinations?

Table 1: “There were 13 patients without MPV values” Can this data be included as the size of the platelets is important for the conclusions drawn.

Flow cytometry: What was the time between sample collection (in 2012) and antigenic assessment? How were the platelets stored prior to study? What was done to prevent platelet activation prior to analysis? What impact did platelet activation have on the results? The authors should refer to the technical and methodological aspects of flow cytometry (e.g. citing MD Linden).

Antigen density has been inferred from the mean fluorescence intensity and forward scatter (size). Can the authors provide data calculated data from a standard using antibody binding capacity of beads. CD36 antigen density was higher in MDS than normal (Table 2). Was this due to platelet activation (“stimulation”)? There was no difference between CD41a antigen density between MDS and normals. This is different from Liu et al (Ref. 28). Why?

Platelet size was measured by MPV (fL), um and Forward Scatter. How do these correlate? How was 7um determined to define “giant platelets” in “blood and BM”? How was the percent giant platelets established? How was the data presented in Figure 5 established?

There are typographical errors (e.g. “powder”, “diagnose” in Discussion)

The majority of references are pre-2010 (and only one in 2020 or later).

The Figures 1 and 2 do not add to the manuscript and could be omitted. Figure is of poor quality

Reviewer #3: The authors describe MFI of platelet markers in MDS showing that, for some antigens, it differs from healthy controls. Large platelets were noted in MDS patients, but not in healthy controls.

The potential role of immunophenotyping in diagnosis of MDS is perhaps over-stated. In most cases it is not required and merely supportive of morphological and genetic findings.

It is stated that platelets have the immunophenotype of megakaryocytes, but there is no citation. Is this proven, or merely assumed?

The study was performed 10 years ago and uses an outdated MDS classification system and does not refer to IPSS-R.

In Methods, please state how the platelet count was determined using Sysmex (impedance or fluorescence).

The age of healthy control donors was substantially lower than in MDS patients. Are there any data on whether the MFI of platelet markers varies with age?

Large platelets are typically seen in thrombocytopenia, so is the presence of large platelets simply a reflection of the severity of thrombocytopenia?

The paragraph that follows Table 1 in Results contains no results - belongs in Introduction/Methods.

One of the problems with platelets in flow cytometry is the risk of clumping, which can sometimes happen with EDTA. How can we be certain that the giant platelets are single platelets and not clumps? The Sysmex MPV did not differ between MDS and healthy controls, which seems to indicate that there is no confirmed difference in size.

The sample size is too small to compare subsets of MDS, so the absence of statistically significant differences probably reflects inadequate power.

Ref #28 seems to have done the same study as the authors, and found no useful differences. Why did the authors repeat the study? Did this study have some advantage that made it worthwhile to re-examine the question?

6. PLOS authors have the option to publish the peer review history of their article (what does this mean?). If published, this will include your full peer review and any attached files.

Reviewer #1: No

Reviewer #2: No

Reviewer #3: No

---

## [Author Response · Author response to Decision Letter 0]

29 Aug 2022

Dear Editor and Reviewers, 

We thank the editor and reviewers for the extensive assessment of the manuscript, and the important and helpful comments and suggestions. We have taken all the remarks into account as described in detail below. We responded to all the comments and revised the manuscript accordingly. We wish the manuscript has improved considerably and suitable for consideration of publication. The detail imformation was described in the file of "Response to Reviewers". If there are any omissions, we will try our best to correct them. Thank you again!

Sincerely

Yi-feng Wu / Tso-fu Wang

---

## [Decision Letter · Decision Letter 1]

25 Oct 2022

PONE-D-22-10816R1Abnormal platelet immunophenotypes and percentage of giant platelets in myelodysplastic syndrome: a pilot studyPLOS ONE

Dear Dr. Wang,

Thank you for submitting your manuscript to PLOS ONE. After careful consideration, we feel that it has merit but does not fully meet PLOS ONE’s publication criteria as it currently stands. Therefore, we invite you to submit a revised version of the manuscript addressing the minor comments from each reviewer.

We look forward to receiving your revised manuscript.

Kind regards,

Daniel Thomas, MD

Academic Editor

PLOS ONE

Journal Requirements:

Additional Editor Comments (if provided):

Most comments have been addressed adequately but the manuscript could still be improved.

Please thoroughly review the grammar with an expert in English.

Please answer the minor comments regarding the Sysmex and giant platelets as per reviewer.

Reviewers' comments:

Reviewer's Responses to Questions

**Comments to the Author**

1. If the authors have adequately addressed your comments raised in a previous round of review and you feel that this manuscript is now acceptable for publication, you may indicate that here to bypass the “Comments to the Author” section, enter your conflict of interest statement in the “Confidential to Editor” section, and submit your "Accept" recommendation.

Reviewer #1: All comments have been addressed

Reviewer #3: (No Response)

2. Is the manuscript technically sound, and do the data support the conclusions?

Reviewer #1: Yes

Reviewer #3: Partly

3. Has the statistical analysis been performed appropriately and rigorously? 

Reviewer #1: Yes

Reviewer #3: Yes

4. Have the authors made all data underlying the findings in their manuscript fully available?

Reviewer #1: Yes

Reviewer #3: Yes

5. Is the manuscript presented in an intelligible fashion and written in standard English?

Reviewer #1: No

Reviewer #3: Yes

6. Review Comments to the Author

Reviewer #1: The authors answered all the comments but, additional corrections are required.

The manuscript requires correction and revision by professional English user.

Reviewer #3: The authors have addressed most of my comments.

The Sysmex instrument has multiple ways to determine the platelet count (e.g. impedance/fluorescence). Please specify which method was used.

Can giant platelets be distinguished from red cell fragments simply by FSC>1000? Was any lineage marker used to confirm?

The paragraph on survival in the discussion could be omitted. The sample size is too small.

7. PLOS authors have the option to publish the peer review history of their article (what does this mean?). If published, this will include your full peer review and any attached files.

Reviewer #1: No

Reviewer #3: No

---

## [Author Response · Author response to Decision Letter 1]

5 Nov 2022

Dear Editor and Reviewers, 

We thank the editor and reviewers for the extensive assessment of the manuscript, and the important and helpful comments and suggestions. We have taken all the remarks into account as described in detail below. We responded to all the comments and revised the manuscript accordingly. We wish the manuscript has improved considerably and suitable for consideration of publication.

Reviewer#1

1. The authors answered all the comments but, additional corrections are required.

2. The manuscript requires correction and revision by professional English user.

Response: 

Thanks for reviewer’s suggestion, we had sent to English editing before submission, and the manuscript had been reviewed with two experts in English again for this revision. The details showed in “Revised Manuscript with Track Changes R2”.

Reviewer#3

1. The authors have addressed most of my comments.

2. The Sysmex instrument has multiple ways to determine the platelet count (e.g. impedance/fluorescence). Please specify which method was used.

3. Can giant platelets be distinguished from red cell fragments simply by FSC>1000? Was any lineage marker used to confirm?

4. The paragraph on survival in the discussion could be omitted. The sample size is too small.

Response: 

1. Thanks for reviewer’s suggestion. 

2. The method with PLT-O (fluorescence) of Sysmex instrument was used for this study. We had added the detail in “Complete blood count test” section as “The PLT-O (fluorescence) channel of Sysmex instrument was used for this study.”

3. We used the platelet specific marker of CD61 for gating platelets for the ratio of “FSC>1000“ which could distinguish from red cell fragments. We added the detail explaining in “Giant platelets in the peripheral blood of MDS patients and healthy controls” section as “On gating platelets by the surface marker of CD61, a subgroup of platelets bearing exceptionally large size (FSC>1000)…”.

4. We had deleted the paragraph about survival analysis in the discussion section.

Thanks for the editor and reviewers about the extensive assessment of the manuscript, and the important and helpful comments and suggestions again.

---

## [Editor Report · Decision Letter 2]

9 Nov 2022

Abnormal platelet immunophenotypes and percentage of giant platelets in myelodysplastic syndrome: a pilot study

PONE-D-22-10816R2

Dear Dr. Wang,

We’re pleased to inform you that your manuscript has been judged scientifically suitable for publication and will be formally accepted for publication once it meets all outstanding technical requirements.

Kind regards,

Daniel Thomas, MD

Academic Editor

PLOS ONE

Additional Editor Comments (optional):

All comments have been adequately addressed
---

## [Editor Report · Acceptance letter]

11 Nov 2022

PONE-D-22-10816R2 

Abnormal platelet immunophenotypes and percentage of giant platelets in myelodysplastic syndrome: a pilot study 

Dear Dr. Wang:

I'm pleased to inform you that your manuscript has been deemed suitable for publication in PLOS ONE. Congratulations! Your manuscript is now with our production department. 

Kind regards, 

on behalf of

Dr. Daniel Thomas 

Academic Editor

PLOS ONE